# Prevalence and Levels of Anti-SARS-CoV-2 Antibodies in the Eswatini Population and Subsequent Severity of the Fourth COVID-19 Epidemic Wave

Portia C. Mutevedzi [1,2,*], Vusie Lokotfwako [3,†], Gaurav Kwatra [1,4,5,6,†], Gugu Maphalala [3], Vicky Baillie [1], Lindiwe Dlamini [3], Senzokuhle Dlamini [3], Fortune Mhlanga [3], Tenelisiwe Dlamini [3], Nhlanhla Nhlabatsi [3], Marta C. Nunes [1,7,8], Simon Zwane [3] and Shabir A. Madhi [1,4]

[1] South African Medical Research Council, Vaccines and Infectious Diseases Analytics Research Unit, Faculty of Health Sciences, University of the Witwatersrand, Johannesburg 2050, South Africa; gaurav.kwatra@wits-vida.org (G.K.); shabir.madhi@wits-vida.org (S.A.M.)
[2] Emory Global Health Institute, Emory University, Atlanta, GA 30322, USA
[3] Eswatini Ministry of Health, Government of Eswatini, Mbabane H100, Eswatini; vusielokza@gmail.com (V.L.); gpmaph@gmail.com (G.M.); lbskonela@gmail.com (L.D.); forthemba@yahoo.com (F.M.); smz1157@gmail.com (S.Z.)
[4] African Leadership in Vaccinology Expertise, Faculty of Health Sciences, University of the Witwatersrand, Johannesburg 2050, South Africa
[5] Department of Clinical Microbiology, Christian Medical College, Vellore 632002, India
[6] Division of Infectious Diseases, Department of Pediatrics, Cincinnati Children's Hospital Medical Center, University of Cincinnati College of Medicine, Cincinnati, OH 45219, USA
[7] Department of Science and Technology/National Research Foundation, South African Research Chair Initiative in Vaccine Preventable Diseases, Faculty of Health Sciences, University of the Witwatersrand, Johannesburg 2050, South Africa
[8] Center of Excellence in Respiratory Pathogens (CERP), Hospices Civils de Lyon (HCL), and Centre International de Recherche en Infectiologie (CIRI), Université Claude Bernard Lyon 1, Inserm U1111, CNRS UMR5308, ENS de Lyon, 6900 Lyon, France
* Correspondence: pmuteve@emory.edu
† These authors contributed equally to this work.

**Abstract:** Background: Seroepidemiology studies are useful for quantifying the magnitude of past infections and estimating the extent of population-based immunity to inform risk mitigation strategies for the future. We report on the only national population-based survey of severe acute respiratory syndrome coronavirus-2 (SARS-CoV-2) immunoglobulin G (IgG) seroprevalence in Eswatini. Methods: The survey was undertaken from 31 August to 30 September 2021, following three earlier waves of coronavirus disease (COVID-19), and preceded the onset of the fourth wave, which was dominated by the Omicron variant of concern. We also report on epidemiological trends of recorded COVID-19 cases and hospitalizations before and after the fourth COVID-19 wave through to March 2022. We evaluated the immunoglobulin G (IgG) seropositivity based on either anti-nucleocapsid (N) or anti-spike (S) antigens. Results: Of 4564 individuals, 58.5% were female, 36.0% were aged 18–50 years, and 863 (18.9%) of adults who were older than 18 years had received at least a single dose of COVID-19 vaccine. Overall, 2769 (60.7%) were seropositive with heterogeneity across sub-regions (53.7%; 95% CI:49.2–58.1 to 68.6%; 95% CI:64.5–72.4), with the highest rates occurring in sub-regions of the Manzini region. Seropositivity was higher in vaccinated individuals (84.5%; 95% CI: 81.9–86.7) compared to unvaccinated individuals (55.1%; 95% CI:53.5–56.7). Amongst unvaccinated individuals, seropositivity was highest in 18–50-year-olds (59.5%;95% CI: 56.9–62.1). Seropositivity was associated with female gender, previous positive SARS-CoV-2 NAAT status and being vaccinated, non-smoking, and being formally employed. We estimated as of 15 September 2021 that there had been 639,475 SARS-CoV-2 infections (95% CI; 620,824–658,003) in Eswatini, which was 25.5-fold greater than the 25,048 COVID-19 cases that had been recorded by then. The national case fatality rate (CFR) based on recorded cases was 4.8%, being 25-fold greater than the infection fatality rate (0.19; 95% CI: 0.18–0.19) based on recorded deaths and extrapolating the force of infection from seroprevalence. Nationally and across all four regions, we report the decoupling of COVID-19 cases

from hospitalisations and deaths, observed as early as during the third wave, which was dominated by the Delta variant compared with earlier waves. Conclusions: We identified that 60.7% of people in Eswatini had been infected by SARS-CoV-2 at least once and before the onset of the Omicron wave in mid-November 2021. Despite a modest uptake of COVID-19 vaccines, the evolution of population immunity from infection has likely contributed to the decoupling of infection and severe COVID-19 in Eswatini.

**Keywords:** seroprevalence; seropositivity; COVID-19; SARS-CoV-2; antibodies; vaccination; anti-nucleocapsid; anti-spike; Immunoglobulin G (IgG)

## 1. Background

Since the onset of the coronavirus-19 disease (COVID-19) pandemic in Africa, Eswatini has experienced four epidemic waves of COVID-19. As of mid-year 2022, when the pandemic had significantly declined, there were 73,060 cases and 1415 deaths recorded in the country [1]. Eswatini, which is landlocked by South Africa and Mozambique, has experienced similar COVID-19 outbreak trajectories compared with South Africa [1]. To date, there are limited severe acute respiratory syndrome coronavirus-2 (SARS-CoV-2) sequencing data from Eswatini, although the circulating variants are expected to be similar to the circulation in South Africa. The Omicron variant of concern (B1.1.529) was first reported on 25 November 2021 in Gauteng Province (South Africa) [2] and subsequently became the dominant variant globally [1].

The Omicron variant was anticipated to be more transmissible compared to earlier variants (Alpha, Beta, Delta) and relatively evasive from neutralizing antibody activity that was induced by the current generation of COVID-19 vaccines or past infection due to the wild type or earlier variants [3]. The heightened transmissibility and antibody-evasiveness of the Omicron variant has been corroborated in studies; however, the memory T-cell responses induced by COVID-19 vaccines or past SARS-CoV-2 infection by earlier variants have been largely conserved against Omicron [4–6].

We undertook a national cross-sectional, population-based survey to determine the seropositivity against SARS-CoV-2 in Eswatini. Furthermore, we assessed sub-regional variability in seropositivity and looked for characteristics associated with seropositivity in the population. The sero-survey took place during the interval period after the third COVID-19 wave had subsided and before the onset of the fourth wave, which was dominated by the Omicron variant of concern and occurred during a period of extremely low COVID-19 vaccine availability and uptake.

## 2. Methods

### 2.1. Study Setting

Eswatini is a small landlocked country in Southern Africa with a population of 1.2 million [7]; the median age is 21 years, with average household size of 4 people and population density of 63 people per square km [7]. The country is demarcated into four health regions constituting eight sub-regions. Seventy-five percent of people in Eswatini live in rural areas, similar to many other African countries [8]. Eswatini reported the first COVID-19 case on 14 March 2020. Testing for SARS-CoV-2 infection is carried out using nucleic acid amplification test (NAAT), which is restricted to symptomatic or highly suspicious cases of COVID-19 [9]. The documented testing rate for SARS-CoV-2 in Eswatini as of 12 March 2022 was 427.7 per 1000 population since the start of the pandemic. The restricted criteria and recent availability of SARS-CoV-2 antigen tests could contribute to high NAAT positivity rates at the peak of outbreak waves [1].

The COVID-19 vaccination programme in Eswatini was initiated in March 2021, with 35,000 doses (3% of the national population) having been vaccinated by May 2021, following

which immunization against COVID-19 was interrupted until August 2021 due to limited vaccine supply.

This survey was conducted from 31 August 2021, after the third COVID-19 outbreak wave had subsided and was completed on 30 September 2021, approximately nine weeks prior to the onset of the fourth COVID-19 wave, which was dominantly caused by the Omicron variant across southern Africa [10]. We obtained raw data on recorded COVID-19 cases and deaths since the onset of the pandemic for Eswatini from the Our World in Data dashboard [1].

### 2.2. Sample Collection and Processing

Dried blood spots (DBSs) were obtained by finger-prick, packed with silica-gel sachets, and transported to Eswatini central laboratory for storage at $-20\ °C$. The specimens were shipped at $-20\ °C$ to the Vaccines and Infectious Diseases Analytics (Wits-VIDA) Research Unit laboratory fortnightly, where elution of antibodies from DBS card specimens was performed as previously described [11–13]. Briefly, one spot was cut from the filter card using a 6 mm hole punch and added to 600 µL of assay buffer. The spot was kept in a shaker at 2–8 °C overnight for elution and the following day was centrifuged at 2000 g for 10 min before analysis. Anti-full-length spike (S) and anti-nucleocapsid (N) immunoglobulin G (IgG) were measured qualitatively on a Luminex platform to determine seropositivity on the DBS sample. Comprehensive details of this methodology have been published [11–13]. Notably, we have previously reported that anti-N IgG is only positive in 39.7% of people with past documented SARS-CoV-2 infection [12]; hence, we used the composite of anti-N or anti-S IgG positivity to determine the seroprevalence.

### 2.3. Sample Size and Sampling

Sample size calculation was based on the WHO population-based survey for SARS-CoV-2 [14]. Assuming SARS-CoV-2 seroprevalence of 20% [15–19] and intra-cluster correlation of 0.33 with design effect of 3.31; precision 0.1; $\alpha = 0.05$; and 7 selected households per cluster, the minimum required sample size was 2086 individuals across the 8 sub-regions.

Sampling was conducted separately for each sub-region to generate seroprevalence estimates at the sub-regional level. Multi-stage sampling methodology was used initially, stratifying the Eswatini 2017 census sampling frame per sub-region and then selecting clusters with the number of clusters per sub-region being proportional to the population size. The number of clusters was inflated by 25% to account for refusals and inaccessible households. Seven households were systematically selected from each cluster by dividing the number of households in the cluster by the required seven households and using that number to define the skip pattern. To ensure sufficient sample size for age-stratified analyses, for each selected household, we pre-identified age groups to be included in the survey.

### 2.4. Data Collection

After obtaining written informed consent, an electronic questionnaire was administered comprising household and individual sections, following which DBS was collected. The household module with household membership listing and socio-demographic characteristics was registered. The individual module included individual-level questions on socio-demographics, previous COVID-19 diagnosis, COVID-19 vaccination, comorbidities, and health-seeking behaviour. Data were collected using password-protected iPads with real-time synchronisation of data into a central database. Where there was no connectivity, data were centrally synchronised at the end of each day.

### 2.5. Statistical Analyses

We estimated seroprevalence as the percentage of individuals testing positive for either anti-S or anti-N IgG and assessed variability by age, gender, vaccination status, and sub-region of residence. We determined the incidence rate ratio (IRR) of and factors associated

with SARS-CoV-2 seropositivity by Poisson regression. Comorbidities included any self-reported hypertension, diabetes, asthma, HIV-positive status, cancer, tuberculosis, stroke, or lung, liver, kidney, or heart disease. We used seroprevalence in the unvaccinated population to calculate national, regional, and sub-regional SARS-CoV-2 infection incidence per 100,000 population. Furthermore, we compared national COVID-19 reported cases with imputed SARS-CoV-2 infection incidence. Using recorded COVID-19 case numbers, we estimated case fatality rates and infection fatality risk (imputing the number of infections based on seroprevalence in unvaccinated individuals) prior to the fourth wave. Finally, we analysed cases, deaths, and hospitalisations due to COVID-19 from the onset of the pandemic through to end December 2021. Survey data analysis techniques in Stata (version 16.1) were used.

*2.6. Ethics*

Ethics approval for the conduction of the survey was obtained from the Eswatini Health and Human Research Review Board (EHHRRB). Written informed consent was obtained from all individuals, and adults consented on behalf of minors.

The authors designed the study; gathered and analysed the data; vouch for the data, the analysis, and adherence to the protocol; and wrote the paper. There was no external support in the writing of the manuscript.

## 3. Results

*3.1. Participants*

We approached 4758 individuals in 2176 households for the sero-survey, of whom 54 (1.1%) individuals declined participation. Dry blood spots were available for 4704 participants. A total of 140 samples (3.0%) could not be linked to an individual, due to data omissions or errors on the sample collection form and were excluded from further analyses. A total of 4564 participants were included in the final analyses; Table 1 shows the number of participants per sub-region and the corresponding seroprevalence.

**Table 1.** Seroprevalence of SARS-CoV-2 anti-spike (anti-S) or anti-nucleocapsid (anti-N) immunoglobulin G (IgG) stratified by sub-region and by vaccination status.

| Sub-Region | * Total Population | Overall Sample Size (%) | [A] Anti-N IgG; n (%); [95% CI] | [B] Anti-S IgG n (%); [95% CI] | [C] Overall Anti-N or Anti-S IgG n (%) [95% CI] | [D] Overall Anti-N or Anti-S IgG n; (%) [ 95% CI]— Unvaccinated | [E] Overall Anti-N or Anti-S IgG n; (%) [95% CI]— Vaccinated |
|---|---|---|---|---|---|---|---|
| Hhohho North | 150,801 | 711 (0.47) | 322 (45.3) [41.7–49.0] | 429 (60.3) [56.7–63.9] | 448 (63.0) [59.4–66.5] | 342/588 (58.2); [54.1–62.1] | 106/123 (86.2); [78.9–91.2] |
| Hhohho South | 187,083 | 648 (0.35) | 204 (31.5) [28.0–35.2] | 327 (50.5) [46.6–54.3] | 352 (54.3) [50.5–58.1] | 254/532 (47.7); [43.5–52.0] | 98/116 (84.5); [76.7–90.0] |
| Hhohho Region Total | 337,884 | 1359 (0.40) | 526 (38.7) [36.1–41.3] | 756 (55.6) [53.0–58.3] | 800 (58.9) [56.2–61.5] | 596/1120 (53.2); [50.3–56.1] | 204/239 (85.4); [80.3–89.3] |
| Lubombo North | 112,330 | 479 (0.43) | 159 (33.2) [29.1–37.5] | 243 (50.7) [46.3–55.2] | 257 (53.7) [49.2–58.1] | 176/378 (46.6); [41.6–51.6] | 81/101 (80.2); [71.3–86.9] |
| Lubombo South | 104,569 | 589 (0.56) | 248 (42.1) [38.2–46.1] | 363 (61.6) [57.6–65.5] | 376 (63.8) [59.9–67.6] | 284/484 (58.7); [54.2–63.0] | 92/105 (87.6); [79.8–92.7] |

**Table 1.** *Cont.*

| Sub-Region | * Total Population | Overall Sample Size (%) | A Anti-N IgG; *n* (%); [95% CI] | B Anti-S IgG *n* (%); [95% CI] | C Overall Anti-N or Anti-S IgG *n* (%) [95% CI] | D Overall Anti-N or Anti-S IgG *n*; (%) [ 95% CI]— Unvaccinated | E Overall Anti-N or Anti-S IgG *n*; (%) [95% CI]— Vaccinated |
|---|---|---|---|---|---|---|---|
| Lubombo Region Total | 216,899 | 1068 (0.49) | 407 (38.1) [35.2–41.1] | 606 (56.7) [53.7–59.7] | 633 (59.3) [56.3–62.2] | 460/862 (53.4); [50.0–56.7] | 173/206 (84.0); [78.3–88.4] |
| Manzini East | 248,446 | 587 (0.24) | 251 (42.8) [38.8–46.8] | 342 (58.3) [54.2–62.2] | 365 (62.2) [58.2–66.0] | 257/462 (55.6); [51.1–60.1] | 108/125 (86.4); [79.2–91.4] |
| Manzini West | 125,757 | 519 (0.41) | 262 (50.5) [46.2–54.8] | 346 (66.7) [62.5–70.6] | 356 (68.6) [64.5–72.4] | 259/406 (63.8); [59.0–68.3] | 97/113 (85.8); [78.1–91.1] |
| Manzini Region Total | 374,203 | 1106 (0.30) | 513 (46.4) [43.5–49.3] | 688 (62.2) [59.3–65.0] | 721 (65.2) [62.3–67.9] | 516/868 (59.4); [56.1–62.7] | 206/238 (86.1); [81.1–90.0] |
| Shiselweni North | 143,190 | 563 (0.39) | 257 (45.6) [41.6–49.8] | 313 (55.6) [51.5–59.7] | 333 (59.1) [55.0–63.1] | 261/473 (55.2); [50.7–59.6] | 72/90 (80.0); [70.5–87.0] |
| Shiselweni South | 87,970 | 468 (0.53) | 192 (41.0) [36.7–45.5] | 264 (56.4) [51.9–60.8] | 282 (60.3) [55.7–64.6] | 207/378 (54.8); [49.7–59.7] | 75/90 (83.3); [74.2–89.7] |
| Shiselweni Region Total | 231,160 | 1031 (0.45) | 449 (43.5) [40.6–46.6] | 577 (56.0) [52.9–59.0] | 615 (59.7) [56.6–62.6] | 468/851 (55.0); 5[1.6–58.3] | 147/180 (81.7); [75.3–86.7] |
| Eswatini National Total | 1,160,146 | 4564 (0.39) | 1895 (41.5) [40.1–43.0] | 2627 (57.6) [56.1–59.0] | 2769 (60.7) [59.2–62.1] | 2040/3701 (55.1); [53.5–56.7] | 729/863 (84.5); [81.9–86.7] |

CI, confidence interval. Seroprevalence was calculated as the number of individuals who were seropositive for anti-S or anti-N IgG divided by the total number of individuals sampled; * population estimates obtained from the Eswatini central statistics office's mid-year population estimates; A overall seroprevalence anti-N IgG; *n* [%; 95% CI]; B overall seroprevalence anti-S IgG; *n* [%; 95% CI]; C overall seroprevalence anti-N or anti-S IgG [%; 95% CI]; D seroprevalence anti-N or anti-S IgG [%; 95% CI] in unvaccinated individuals of all age groups; E seroprevalence anti-N or anti-S IgG [%; 95% CI] in vaccinated adults ≥ 18 years.

Overall, 19% (863/4564) of the participants reported having received at least a single COVID-19 vaccine dose, including 33.1% (863/2608) of people ≥ 18 years who were eligible to receive a COVID-19 vaccine at the time, of whom 88.6% (*n* = 765) had a vaccine card. The seroprevalence was similar in those with (84.7; 95% CI, 82.0–87.1) and without (82.8; 95% CI 74.4–88.9) vaccination cards who reportedly had received a vaccine; hence, self-reported vaccination status was considered as being vaccinated. Demographic and household characteristics, the prevalence of known underlying medical conditions, self-reported HIV status, and vaccination status are reported in Table 2.

**Table 2.** Risk factors for seropositivity in Eswatini, adjusted for gender, age, and vaccination status.

| | *n* (%) | Seroprevalence % *n*; (%) [95 CI] | uaIRR [95 CI] | aIRR [95 CI] |
|---|---|---|---|---|
| Gender: Male | 1895 (41.5) | 1046 (55.2) [52.9–57.4] | Ref | |
| Female | 2669 (58.5) | 1723 (64.6) [62.7–66.3] | 1.17 [1.11–1.23] | 1.12 [1.06–1.18] |

**Table 2.** *Cont.*

| | *n* (%) | Seroprevalence % *n*; (%) [95 CI] | uaIRR [95 CI] | aIRR [95 CI] |
|---|---|---|---|---|
| * Vaccination status: Unvaccinated | 3701 (81.1) | 2040 (55.1) [53.5–56.7] | Not evaluated | Not evaluated |
| Vaccinated | 863 (18.9) | 729 (84.5) [81.9–86.7] | | |
| Vaccination status by age: <5 years, unvaccinated | 671 (14.7) | 319 (47.5) [43.8–51.3] | 0.80 [0.73–0.87] | 0.75 [0.67–0.85] |
| 5–11 years, unvaccinated | 751 (16.5) | 366 (48.7) [45.2–52.3] | 0.82 [0.75–0.89] | 0.79 [0.72–0.87] |
| 12–17 years, unvaccinated | 534 (11.7) | 333 (62.4) [58.2–66.4] | 1.05 [0.97–1.13] | 1.00 |
| 18–50 years, unvaccinated | 1342 (29.4) | 799 (59.5) [56.9–62.1] | Ref | |
| 18–50 years, vaccinated | 300 (6.6) | 262 (87.3) [83.1–90.6] | 1.47 [1.38–1.56] | 1.40 [1.31–1.49] |
| 51–65 years, unvaccinated | 271 (5.9) | 161 (59.4) [53.5–65.1] | 1.00 [0.90–1.11] | 0.98 [0.88–1.09] |
| 51–65 years, vaccinated | 315 (6.9) | 267 (84.8) [80.4–88.3] | 1.42 [1.33–1.52] | 1.37 [1.28–1.47] |
| >65 years, unvaccinated | 132 (2.9) | 62 (47.0) [38.6–55.5] | 0.79 [0.65–0.95] | 0.80 [0.66–0.96] |
| >65 years, vaccinated | 248 (5.4) | 200 (80.7) [75.3–85.1] | 1.35 [1.26–1.46] | 1.35 [1.25–1.47] |
| Reported previous COVID-19 test: Never tested | 3813 (83.6) | 2210 (58.0) [56.4–59.5] | Ref | |
| Tested positive | 156 (3.4) | 146 (93.6) [88.5–96.5] | 1.61 [1.54–1.70] | 1.44 [1.36–1.53] |
| Tested negative | 595 (13.0) | 413 (69.4) [65.6–73.0] | 1.20 [1.13–1.27] | 1.04 [0.98–1.10] |
| ¥ Residential type: Formal stand-alone house | 2667 (58.5) | 1551 (58.2) [56.3–60.0] | Ref | |
| Informal or traditional dwelling | 1690 (37.1) | 1097 (64.9) [62.6–67.2] | 1.12 [1.06–1.17] | 1.00 [0.95–1.05] |
| Block of flats/high rise buildings | 204 (4.5) | 120 (58.8) [51.9–65.4] | 1.01 [0.90–1.14] | 0.94 [0.84–1.06] |
| Occupation: Unemployed | 2351 (51.5) | 1418 (60.3) [60.3; 58.3–62.3] | Ref | |
| Student | 1444 (31.6) | 789 (54.6) [52.1–57.2] | 0.91 [0.86–0.96] | 0.98 [0.91–1.06] |
| Formally employed | 541 (11.9) | 411 (76.0) [72.2–79.4] | 1.26 [1.19–1.33] | 1.12 [1.05–1.19] |
| Informally employed | 228 (5.0) | 151 (66.2) [59.8–72.1] | 1.10 [1.00–1.21] | 1.03 [0.93–1.13] |
| † Smoking: Smoker | 198 (4.3) | 100 (50.5) [43.6–57.4] | Ref | |
| Non-smoker | 4366 (95.7) | 2669 (61.6) [59.7–62.6] | 1.21 [1.05–1.39] | 1.21 [1.05–1.38] |
| Multiple morbidity: None | 1582 (34.7) | 1031 (65.2) [62.8–67.5] | Ref | |

**Table 2.** *Cont.*

|  | *n*<br>(%) | Seroprevalence %<br>*n*; (%) [95 CI] | uaIRR<br>[95 CI] | aIRR<br>[95 CI] |
|---|---|---|---|---|
| 1 or more | 1026<br>(22.5) | 720 (70.2)<br>[67.3–72.9] | 1.08<br>[1.02–1.14] | 0.99<br>[0.94–1.05] |
| Under 18 years | 1956<br>(42.9) | 1018 (52.0)<br>[49.8–54.3] | Not evaluated |  |
| Region: Manzini | 1106<br>(24.2) | 721 (65.2)<br>[62.3–67.9] | Ref |  |
| Hhohho | 1359<br>(29.8) | 800 (58.9)<br>[56.2–61.5] | 0.90<br>[0.85–0.96] | 0.92<br>[0.87–0.98] |
| Lubombo | 1068<br>(23.4) | 633 (59.3)<br>[56.3–62.2] | 0.91<br>[0.85–0.97] | 0.93<br>[0.87–0.99] |
| Shiselweni | 1031<br>(22.6) | 615 (59.7)<br>[56.6–62.6] | 0.92<br>[0.86–0.98] | 0.95<br>[0.89–1.01] |

uaIRR, unadjusted incidence rate ratio. aIRR, adjusted incidence rate ration. CI, confidence interval. * Age and vaccination status were not individually included in the adjusted regression model; instead, we introduced an interaction term between age and vaccination status to account for the differences in seroprevalence by vaccination status across the different age categories. † Smoking status was restricted to individuals aged >18 years in the univariable analyses. We determined incidence rate ratio of SARS-CoV-2 seropositivity by Poisson regression. Unadjusted and adjusted incidence rate ratios are presented with 95% confidence intervals (CI) in brackets. We used the national census classification to define dwelling types. ¥ Three individuals did not have type of dwelling captured. Ref: reference/comparison group.

### 3.2. Seroprevalence

The overall seropositivity rate (anti-S or anti-N IgG) for Eswatini was 60.7% (95% CI: 59.2–62.1), ranging between 58.9% and 65.2% across regions (Table 1). Manzini had the highest seroprevalence of 65.2% (95% CI: 62.3–67.9). The seropositivity was lower in Hhohho (58.9%; aIRR, 0.92; 95% CI, 0.87–0.98) and Lubombo (59.3%; aIRR 0.93; 95% CI, 0.87–0.99) compared with the Manzini region (Table 2). The seroprevalence was more heterogenous between sub-districts, ranging from 53.7% (95% CI: 49.2–58.1) in Lubombo North to 68.6% (95% CI: 64.5–72.4) in Manzini West (Table 1).

In people who were unvaccinated against COVID-19, the overall seropositivity was 55.1% (95% CI: 53.5–56.7), which is lower than the 84.5% (95% CI: 81.9–86.7) in vaccinated individuals aged 18 years and above. Across all regions and sub-regions, the seroprevalence was lower in unvaccinated than in vaccinated individuals, ranging from 46.6% (95% CI: 41.6–51.6) in Lubombo North to 63.8% (95% CI: 59.0–68.3) in Manzini West (Table 1).

The overall anti-N IgG seropositivity was 41.2% (1523/3701) in the COVID-19 unvaccinated individuals, including 46.5% (812/1745) in people who were older than 18 years and being similar between the 18- to 50-year-old (45.9%; 95% CI: 43.2–48.6) and >50-year-old (48.6%; 95% CI: 43.8–53.5) age groups. Of those vaccinated, 43.1% (372/863; 95% CI: 39.8–46.4) were positive for anti-N IgG, which was similar between the 18–50 year—39.7% (119/300; 95% CI: 34.3–45.3)—and >50 year—44.9% (253/563; 95% CI: 40.9–49.1)—age groups.

The seropositivity was similar across all the adult age groups who had been vaccinated, ranging from 80.6% in the >65-year age group to 87.3% in the 18 to 50 year age group (Table 2). In the multivariable analysis, among the unvaccinated, compared with the seropositivity in the 18- to 50-year-old age group (59.5%) as a reference group, the seropositivity was lower in the under-5 (47.5%; aIRR: 0.75; 95% CI: 0.67–0.85), 5–11 year (48.7%; aIRR: 0.79; 95% CI: 0.72–0.87), and >65 year (47.0%; aIRR: 0.80; 95% CI: 0.66–0.96) age groups. Also, compared with the 18- to 50-year-old age group (59.5%) who were unvaccinated, the seropositivity was higher across all age groups who were eligible to be vaccinated (Table 2).

In a further multivariable analysis, females were more likely to be seropositive, with an adjusted incidence rate ratio (aIRR) of 1.12 (95% CI: 1.06–1.18) compared to males.

Individuals who reported having previously tested positive for SARS-CoV-2 infection were at higher risk of being seropositive (93.6%) than those who had never tested (58.0%; aIRR 1.44, 95% CI: 1.36–1.53). Compared with unemployed individuals, individuals in formal employment had a higher risk of being seropositive (66.2%; aIRR, 1.12, 95% CI 1.05–1.19). Compared to smokers (50.5%; 95% CI: 43.6–57.4), the seropositivity was higher in non-smokers (61.1%; 95% CI: 59.7–62.6) with an aIRR of 1.21 (95% CI: 1.05–1.38) (Table 2).

### 3.3. Calculated SARS-CoV-2 Infections

We used the seroprevalence (55.1%) in unvaccinated individuals (*n* = 3701) to impute the population rate of SARS-CoV-2 infections at the sub-regional, regional, and national levels. We estimated that as of 15 September 2021, there had been 639,475 SARS-CoV-2 infections (95% CI; 620,824–658,003) in Eswatini, compared with only 25,048 recorded COVID-19 cases by then. The difference in recorded COVID-19 cases versus calculated SARS-CoV-2 infections was largest in Hhohho North (73.0-fold difference) and smallest in Shiselweni South (12.4-fold). Even within regions, the rate of under-reporting was variable; for example, in Hhohho region, there was a 73.0-fold difference in reported cases vss calculated infections, whilst in Hhohho South the fold-difference was only 17.6-fold.

Due to the 25.5-fold under-reporting of COVID-19 cases, the national calculated incidence for SARS-CoV-2 infections (per 100,000 population) was 55,120 compared with 2159 for recorded cases (Table 3). The calculated SARS-CoV-2 infection incidence (per 100,000 population) was highest in the Manzini region of Eswatini (*n* = 59,447), whilst it was similar in other regions (*n* = 53,214 to 54,994). The calculated incidence (per 100,000) of infections differed by sub-region, ranging from 47,744 in Hhohho South to 63,793 in Manzini West (Table 3).

**Table 3.** Demographics of sampling area in sero-survey, calculated number of COVID-19 cases based on seroprevalence in unvaccinated individuals compared to reported cases.

| Sub-Region | n | A Seroprevalence | B Imputed SARS-CoV-2 Infections | C Imputed SARS-CoV-2 Infection Incidence Per 100,000 | D Documented COVID-19 Cases | E Documented COVID-19 Cases Per 100,000 |
|---|---|---|---|---|---|---|
| Hhohho North | 588 | 342 (58.2) [54.1–62.1] | 87,711 [81,628–93,633] | 58,163 [54,130–62,090] | 1202 | 797 |
| Hhohho South | 532 | 254 (47.7) [43.5–52.0] | 89,322 [81,428–97,276] | 47,744 [43,525–51,996] | 5083 | 2717 |
| Region Total | 1120 | 596 (53.2) [50.3–56.1] | 179,803 [169,900–189,631] | 53,214 [50,283–56,123] | 6285 | 1860 |
| Lubombo North | 378 | 176 (46.6) [41.6–51.6] | 52,302 [46,709–57,972] | 46,561 [41,582–51,609] | 3262 | 2904 |
| Lubombo South | 484 | 284 (58.7) [54.2–63.0] | 61,359 [56,710–65,864] | 58,678 [54,232–62,987] | 1531 | 1464 |
| Region Total | 862 | 460 (53.4) [50.0–56.7] | 115,747 [108,499–122,929] | 53,364 [50,023–56,676] | 4793 | 2210 |
| Manzini East | 462 | 257 (55.6) [51.1–60.1] | 138,205 [126,860–149,318] | 55,628 [51,062–60,101] | 6377 | 2567 |
| Manzini West | 406 | 259 (63.8) [59.0–68.3] | 80,224 [74,196–85,926] | 63,793 [58,999–68,327] | 1595 | 1268 |
| Region Total | 868 | 516 (59.4) [56.1–62.7] | 222,452 [210,086–234,507] | 59,447 [56,142–62,668] | 7972 | 2130 |
| Shiselweni North | 473 | 261 (55.2) [50.7–59.6] | 79,012 [72,549–85,354] | 55,180 [50,666–59,609] | 2127 | 1485 |

**Table 3.** *Cont.*

| Sub-Region | *n* | A Seroprevalence | B Imputed SARS-CoV-2 Infections | C Imputed SARS-CoV-2 Infection Incidence Per 100,000 | D Documented COVID-19 Cases | E Documented COVID-19 Cases Per 100,000 |
|---|---|---|---|---|---|---|
| Shiselweni South | 378 | 207 (54.8) [49.7–59.7] | 48,174 [43,731–52,532] | 54,762 [49,711–59,716] | 3871 | 4400 |
| Region Total | 851 | 468 (55.0) [51.6–58.3] | 127,124 [119,355–134,790] | 54,994 [51,633–58,310] | 5998 | 2595 |
| Eswatini | 3701 | 2040 (55.1) [53.5–56.7] | 639,475 [620,824–658,003] | 55,120 [53,513–56,717] | 25,048 | 2159 |

* Population estimates obtained from the Eswatini central statistics office mid-year population estimates; population density per region: Hhohho—89; Lubombo—36; Manzini—87; Shiselweni—54; Eswatini country—63; A seroprevalence (N+ or S+) N; (%) [95% CI]; B imputed SARS-CoV-2 infections based on seroprevalence as of 30 September 2021 [95% CI]; C imputed SARS-CoV-2 infection incidence per 100,000 population as of 30 September 2021 (based on seroprevalence data) [95% CI]; D documented COVID-19 cases as of 15 September 2021; E documented COVID-19 cases per 100,000 population as of 15 September 2021.

### 3.4. COVID-19 Fatality Rates

Overall, 1199 COVID-19 deaths were recorded from 25,048 cases, through to 15 September 2021, giving a case fatality risk (CFR) of 4.8% and a mortality rate of 103.3 per 100,000 population. There was significant heterogeneity in the CFR at the regional and sub-regional levels, ranging from 12.5% in Hhohho North to 1.9% in Shiselweni South. There was no correlation between the recorded CFR and mortality rates per 100,000 population. Hhohho North, the sub-region with the highest CFR (12.5%), was amongst the lowest in terms of mortality rates (99.5 per 100,000 population), whilst Manzini East (CFR 6.0) had the highest mortality rate (153.4). The large variance between reported cases and imputed infections resulted in the national CFR (4.8%) being 25-fold greater than the infection fatality risk (0.19%; 95% CI:0.18–0.19), albeit both were calculated based on recorded deaths (Table 4).

**Table 4.** Adjusted SARS-CoV-2 infection fatality rate based on reported COVID-19 deaths and on calculated SARS-CoV-2 infections prior to widespread vaccine implementation.

| Region | Sub-Region | Total Population Size | A COVID-19 | B Mortality Rate COVID-19 | C COVID-19 CFR COVID-19 | D Imputed IFR |
|---|---|---|---|---|---|---|
| Hhohho | Hhohho North | 150,801 | 150 | 99.5 | 12.5 | 0.17 [0.16–0.18] |
| | Hhohho South | 187,083 | 215 | 114.9 | 4.2 | 0.24 [0.22–0.26] |
| | Region Total | 337,884 | 365 | 108.0 | 5.8 | 0.20 [0.19–0.21] |
| Lubombo | Lubombo North | 112,330 | 64 | 57.0 | 2.0 | 0.12 [0.11–0.14] |
| | Lubombo South | 104,569 | 48 | 45.9 | 3.1 | 0.08 [0.07–0.08] |
| | Region Total | 216,899 | 112 | 51.6 | 2.3 | 0.10 [0.09–0.10] |
| Manzini | Manzini East | 248,446 | 381 | 153.4 | 6.0 | 0.28 [0.26–0.30] |
| | Manzini West | 125,757 | 139 | 110.5 | 8.7 | 0.17 [0.16–0.19] |
| | Region Total | 374,203 | 520 | 139.0 | 6.5 | 0.23 [0.22–0.25] |
| Shiselweni | Shiselweni North | 143,190 | 127 | 88.7 | 6.0 | 0.16 [0.15–0.18] |
| | Shiselweni South | 87,970 | 75 | 85.3 | 1.9 | 0.16 [0.14–0.17] |
| | Region Total | 231,160 | 202 | 87.4 | 3.4 | 0.16 [0.15–0.17] |
| National Total | Eswatini | 1,160,146 | 1199 | 103.3 | 4.8 | 0.19 [0.18–0.19] |

A Documented COVID-19 deaths as of 30 September 2021; B mortality rate per 100,000 population based on documented COVID-19 deaths; C COVID-19 case fatality risk (CFR) based on documented COVID-19 deaths (%); D imputed infection fatality rate (IFR) based on community-based seroprevalence (%) [95% CI] but using recorded deaths.

*3.5. Decoupling of COVID-19 Cases, Hospitalizations, and Deaths*

The trajectory of each of the four COVID-19 waves in Eswatini is shown in Figure 1 and in Supplementary Table S1. Nationally, the highest number of cases was reported during the fourth wave (29 November 2021 to 14 April 2022), despite the high seroprevalence reported here before the onset of the fourth wave. During the first epidemic wave, which happened between 24 February 2020 and 12 October 2020, there were 6008 reported COVID-19 cases (11.2% of total cumulative cases) compared to 26,583 (49.7% of total cumulative cases) reported during the fourth wave between the 29 November 2021 and 14 April 2022. The cumulative case rate per 100,000 population was 517.9, 637.8, 1163.0, and 2291.4 for waves 1–4, respectively. The increase in case rates was decoupled from the case severity, as shown by 1 in 34 COVID-19 cases resulting in death in the first wave, increasing to 1 in 14 and 1 in 26 in the second and third wave and significantly declining to 1 in 159 in the fourth wave. As such, the case fatality risk was initially at 3.0% in the first wave, and increased to 7.2% in the second wave, before declining to 0.6% in the fourth wave, as shown in Figure 1 and Supplementary Table S1.

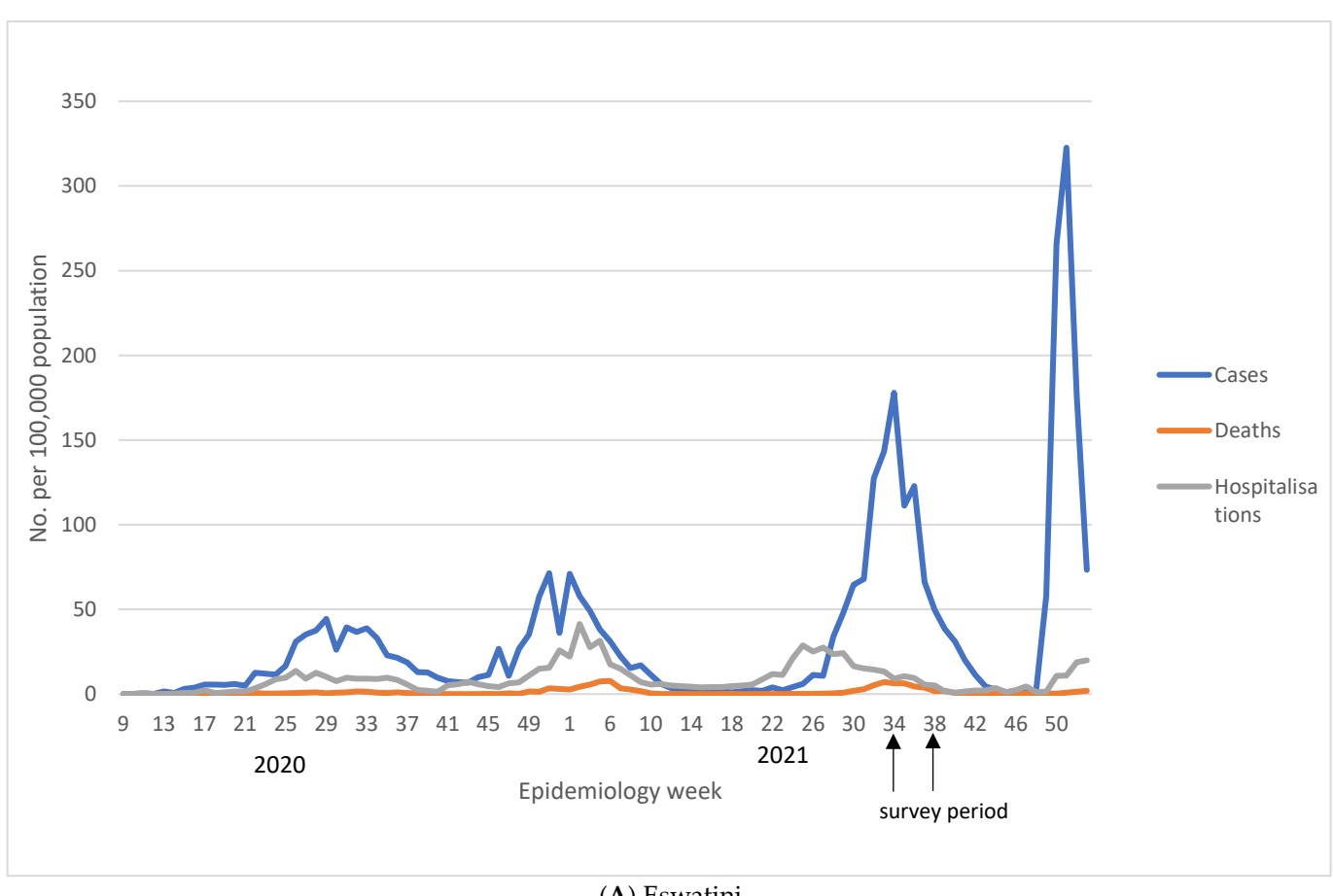

(**A**) Eswatini

**Figure 1.** *Cont.*

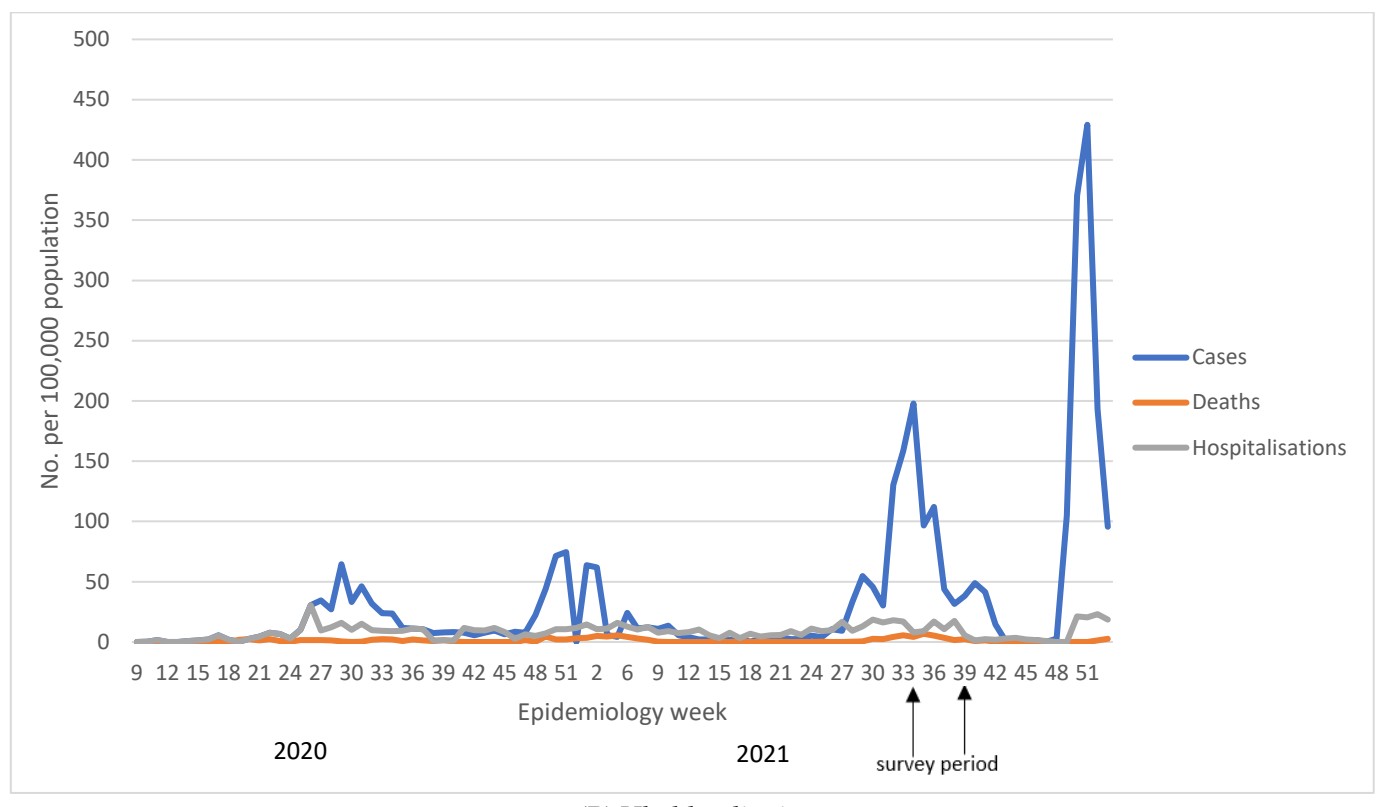

(**B**) Hhohho district

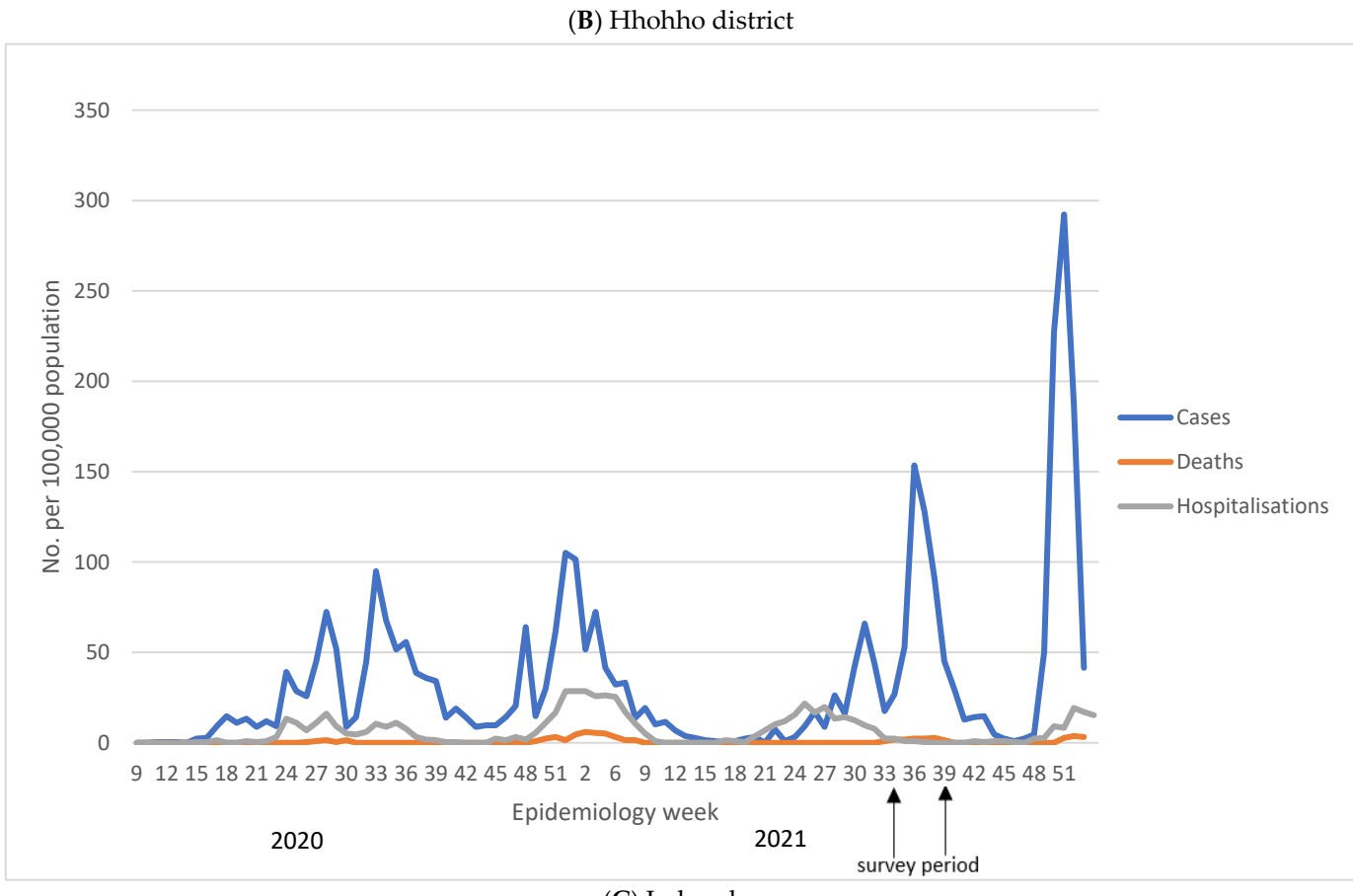

(**C**) Lubombo

**Figure 1.** *Cont.*

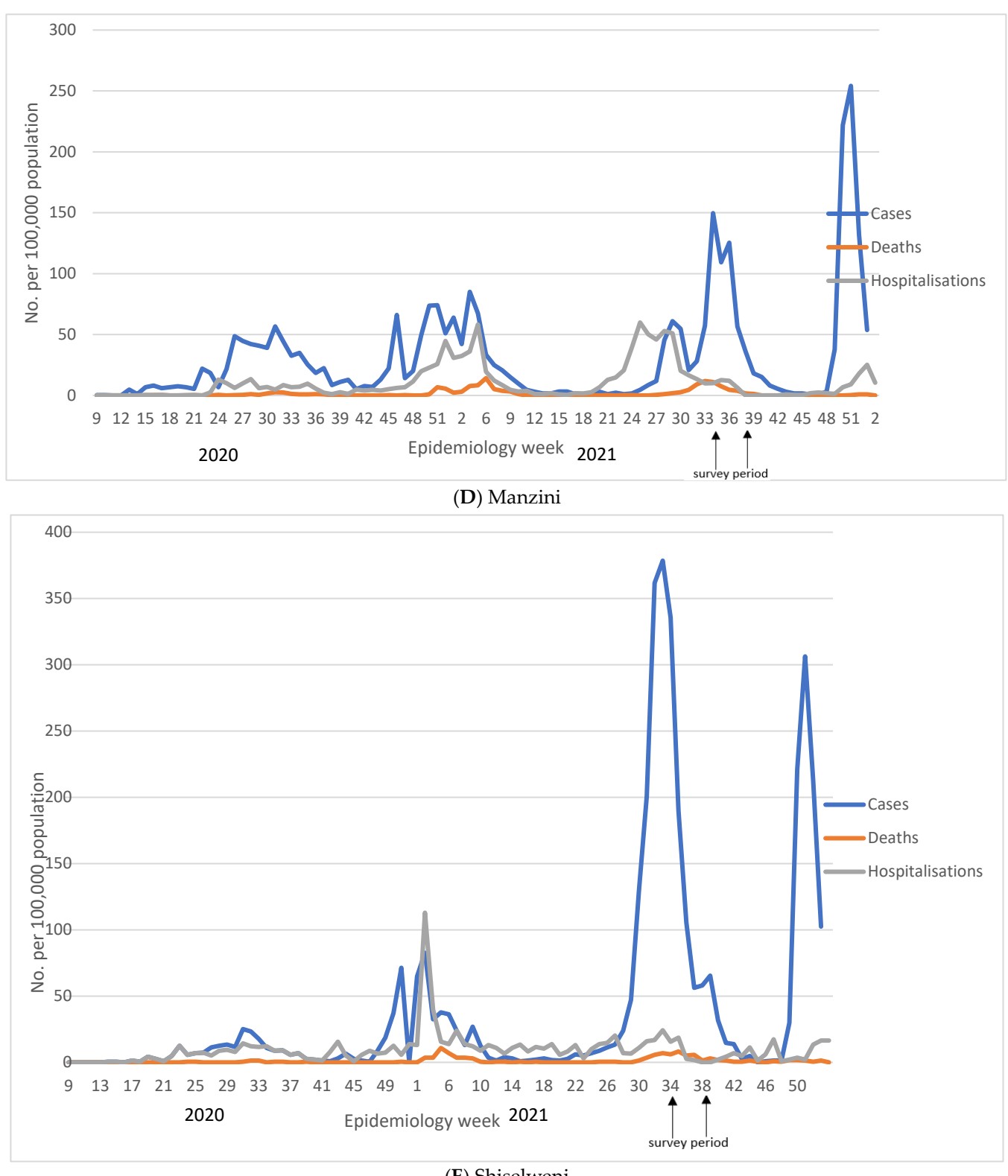

**Figure 1.** Reported COVID-19 cases, hospitalisations, and deaths in Eswatini at national (**A**) and regional (**B–E**) level. We show number of COVID-19 cases, deaths, and hospitalisations reported per week obtained from Eswatini Ministry of Health. Figure (**A**) reports national figures, whilst figures (**B–E**) represent regional numbers. For all regions, except for Shiselweni, each subsequent wave had higher number of cases compared to the previous wave. The fourth wave was associated with the lowest number of hospitalisations and deaths.

## 4. Discussion

We report on the only seroprevalence survey in Eswatini since the onset of the COVID-19 pandemic, adding to the few other population-based seroprevalence studies in Africa [20], and report a high seroprevalence of 61% in Eswatini before the onset of the fourth COVID-19 wave. The high seropositivity was evident across all regions, including as high as 69% in the capital sub-regions (Manzini East and West), which was dominantly due to prior SARS-CoV-2 infection, as indicated by only 18.9% of people in Eswatini having received at least a single dose of COVID-19 vaccine at the time of the survey, and 55.1% of COVID-19 unvaccinated individuals being seropositive. The high seropositivity in Eswatini, a predominantly rural African setting, among unvaccinated individuals is corroborated by the results from a household transmission study on SARS-CoV-2 in rural and urban sites in South Africa. In the South African study, the attack rate over a 13-month follow-up period up until August 2021, by when only 4% of the study population had been vaccinated, were 68.4% and 57.2% in the urban and rural setting, respectively [21].

Eswatini has a homogenous population with most of the country being predominantly rural [8]; in age-, gender- and vaccination status-adjusted analyses, seropositivity was not associated with residential type. This may have led to little variability in the seroprevalence between regions compared to previous studies that reported significant variability in the seroprevalence between regions [13,16,19,22]. Eswatini's most densely populated region has a population density (people per km$^2$) of 89, with the lowest being 36. Manzini, the capital with a population density 87, had the highest seroprevalence; however, in Hhohho, despite having a high population density of 89, the seroprevalence was similar to that of Shiselweni with a population density of 36.

The 13-fold difference between the imputed numbers of SARS-CoV-2 infection compared with recorded deaths highlights the challenges of being able to adequately quantify the burden of COVID-19 in African settings such as Eswatini, where although the testing rates of 428 per 1000 as of March 2022 were higher than in most other African countries, they are significantly lower than those of high-income countries such as the United Kingdom (3792) and USA (2490) [1,20]. Our study shows that based on the overall anti-S or anti-N seroprevalence, there was an estimated 639,475 SARS-CoV-2 cases in Eswatini compared to only 25,048 reported cases. A meta-analysis of 54 seroprevalence reports, including 151 distinct studies in Africa, reported that the ratio of seroprevalence to cumulative incidence of reported cases can be as high as 958:1 [23]. Of the 3409 seroprevalence studies reported on the SeroTracker as of 14 April 2022, all reports show gross under-reporting of SARS-CoV-2 infections using reported vs. calculated cases based on seroprevalence, with under-reporting being more evident in resource-limited settings with stringent COVID-19 testing algorithms [20]. Similarly, household transmission studies have also shown large gaps in prevalence when using RT-PCR positive cases (1.7%) compared to serology positive cases (62%) [21]. A couple of reasons may have contributed to this finding: (1) limited access to testing services and stringent testing algorithms result in testing of severe cases; and (2) high rates of asymptomatic cases that would not be aware of infection, thus neither seeking health care, nor being tested. African studies from similar contexts have reported as many as 85% of SARS-CoV-2 cases being asymptomatic despite active symptom evaluation twice a week [21].

Under-documentation of COVID-19 cases could inadvertently lead to over-estimation of the IFR, whilst conversely, under-ascertainment of COVID-19 deaths could underestimate the IFR. Based on documented cases and deaths, the CFR of COVID-19 in Eswatini was 20-fold higher than the imputed IFR (4.79 vs. 0.19%) based on recorded deaths and sero-surveys of imputed SARS-CoV-2 infections. Nevertheless, recent data indicate that the recorded COVID-19 deaths possibly underestimate COVID-19-attributable deaths based on excess mortality estimates by a factor of 3, with the difference being greatest in low- and middle-income countries [24]. In South Africa, the calculated COVID-19-attributable mortality rates using excess mortality estimates were three-fold higher compared with recorded COVID-19 deaths, being five-fold higher in the more rural non-metropolitan

provinces [13,25]. A recent modelling exercise by the COVID-19 Excess Mortality Collaborators estimated that the ratio of excess mortality rates to reported COVID-19 mortality rates could be as high as 10-fold [24], thus inferring that the IFR and CFR could be 10-fold higher than we estimate here.

An important finding of this study is that of a high number of cases which transpired during the fourth COVID-19 wave, which is attributable to the Omicron variant, after the sero-survey had been conducted, despite 60.7% of the population being seropositive for SARS-CoV-2. The relatively high case rate during the Omicron wave compared with the first and second COVID-19 waves could be due to the greater transmissibility of Omicron compared with the wild-type virus or earlier variants that circulated. Also, the relatively high case rate, despite 60.7% of the population being seropositive, is likely due to re-infections in previously infected individuals, as well as breakthrough infections in vaccinated individuals being common with Omicron due to its relative antibody evasiveness from immunity that has been induced by earlier variants, as well as the current generation of COVID-19 vaccines [26,27]. Nevertheless, caution needs to be exercised in head-to-head comparisons during the course of the pandemic, and particularly for cases recorded which could have been affected in terms of ascertainment based on the rate of testing for COVID-19 as well as the threshold that was used to test individuals.

Our study indicates an increase in cases with each subsequent wave post the Wuhan-Hu-1 and a similar increase in recorded COVID-19 hospitalisations and deaths between the first and second wave. However, there is evidence that as early as during the third delta wave, despite the high number of cases, the incidence of recorded COVID-19 hospitalisations and deaths was lower compared with the earlier waves that were caused by the wild-type virus and beta variant of concern. This is possibly attributable to a force of infection prior to the third wave which was dominated by the Delta variant of concern, which may have been sub-optimal in protecting against infection but adequate for protecting against severe COVID-19 illness and death [28–31]. In South Africa, a similar decoupling of cases from hospitalisations and deaths was reported and attributed to high population immunity, mainly due to infection [12], but this was only observed with the fourth wave, dominated by the Omicron variant of concern, whereas the preceding wave that was caused by the Delta variant was characterised as the most severe since the onset of the pandemic. The disconnect between infection or vaccines being able to bring about a disproportionately greater reduction in severe COVID-19 compared with the effectiveness thereof against infection and mild COVID-19 for variants such as Omicron is attributed to T-cell-induced immunity, from either mechanism of immunity being relatively unaffected by the mutations being harboured, as opposed to the variant being evasive to neutralising antibody activity, which is considered to be required to protect against infection and mild COVID-19 [2,4,6,30,31].

Comparing our study's population demographics with those from the Eswatini population bureau, our study is largely representative of the Eswatini population in terms of age (median 22 years for both Eswatini and survey population) and proportion of urban population. Our survey sample included 59% women, compared to the background population having 51% women. Our finding of higher odds of seropositivity in women may have resulted in an overestimated seroprevalence. The Eswatini employment rate of 37% is higher than the 25% in our study, which would have underestimated seroprevalence, since the odds of seropositivity were higher among the employed. Even though every effort was made through the sampling strategy to ensure representativeness, there may have been residual biases reducing the generalization of results to the entire Eswatini population. Additionally, behavioural differences and other socio-economic factors not captured in this study may have confounded the higher seropositivity that was observed in women and amongst those who are employed. Antibody tests have limitations [32], and waning of antibodies in individuals who were previously infected would have inevitably led to an underestimated seroprevalence. Since no confirmatory virus neutralization test was performed, anti-N antibodies may be cross-reactive, resulting in false positive results, thus

overestimating seroprevalence. The comparison of seroprevalence studies from different settings should be carried out cautiously, taking into consideration the timing of the survey in relation to the outbreak trajectory. Generally, studies conducted later in the outbreak have higher seropositivity [20].

Despite a high force of infection that has taken place in Eswatini before the onset of the Omicron wave and a seropositivity of 59.4% and 47.0% in the 51–65 year and >65 year age groups who had not been vaccinated, there remains a high percentage of people of >50 years of age who are susceptible to SARS-CoV-2 infection and at the greatest risk of severe COVID-19 illness. The higher antibody titres in vaccinated compared to unvaccinated individuals in our study shows that a role remains for ongoing immunization against COVID-19, and particularly for people who are older than 50 years who contribute to >80% of all COVID-19 deaths, even in African settings [33]. Also, with extensive infection-induced immunity, COVID-19 vaccination would result in hybrid immunity, which purportedly is able to overcome even the relative antibody evasiveness of Omicron compared with infection- or vaccine-only-induced immunity.

**Supplementary Materials:** The following supporting information can be downloaded at https://www.mdpi.com/article/10.3390/covid4030021/s1: Table S1: Cumulative reported COVID-19 cases, hospitalizations and recorded deaths in Eswatini by region and COVID-19 wave.

**Author Contributions:** Conceptualization, P.C.M., V.L., G.K., V.B., S.Z. and S.A.M.; methodology, P.C.M., G.K., G.M., V.B., F.M. and S.A.M.; software, S.D.; validation, P.C.M., V.L., G.K., G.M., V.B., L.D. and F.M.; formal analysis, P.C.M., G.K., S.D. and V.B.; investigation, G.K., V.B. and L.D.; data curation, L.D., S.D. and T.D.; writing—original draft preparation, P.C.M. and G.K.; writing—review and editing, P.C.M., V.L., G.K., G.M., V.B., L.D., S.D., F.M., T.D., N.N., M.C.N., S.Z. and S.A.M.; visualization, P.C.M.; supervision, P.C.M., V.L., L.D., F.M. and T.D.; project administration, V.L., L.D. and T.D.; funding acquisition, P.C.M. and S.A.M. All authors have read and agreed to the published version of the manuscript.

**Funding:** This research was funded by Astrazeneca grant number [D8111C00012 - ESR-21-21294].

**Institutional Review Board Statement:** Ethics approval for the conduct of the survey was obtained from the Eswatini Health and Human Research Review Board (EHHRRB). Approval number EHHRRB032/2021.

**Informed Consent Statement:** Written informed consent was obtained from all subjects involved in the study.

**Data Availability Statement:** Data are available at www.wits-vida.org; requests for data sharing should be directed to Shabir Madhi (shabir.madhi@wits.ac.za).

**Conflicts of Interest:** The authors declare no conflict of interest.

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
