# Peer review of "Prevalence and Levels of Anti-SARS-CoV-2 Antibodies in the Eswatini Population and Subsequent Severity of the Fourth COVID-19 Epidemic Wave"

_covid, doi:10.3390/covid4030021_

Round 1

Reviewer 1 Report

Comments and Suggestions for Authors

In this study, they demonstrated that 60.7% of the population in Eswatini had been infected with SARS-CoV-2 at least once before the onset of the Omicron wave in mid-November 2021. Despite the modest uptake of COVID-19 vaccines, the evolution of population immunity from infection has likely contributed to the decoupling of infection and severe COVID-19 in Eswatini. These findings emphasize the importance of sero-epidemiology studies in understanding the extent of past infections and informing future risk mitigation strategies.

While efforts were made to ensure representativeness, it is important to note that the study's findings may not be fully generalizable to the entire population of Eswatini. The sampling methodology utilized may have introduced biases, and there may be subgroups or regions that were not adequately represented in the study sample.

-The study acknowledges higher odds of seropositivity in women and among the employed. However, it is essential to consider potential confounding factors that may influence these associations. Factors such as occupation, socioeconomic status, and behavioral differences could contribute to the observed differences in seropositivity rates.

-Antibody tests have their limitations, including the possibility of false negatives or false positives. Additionally, while the study acknowledges the waning of antibodies in individuals previously infected, the extent of this waning and its impact on seroprevalence estimation may not be fully captured.

-The study notes that studies conducted later in the outbreak tend to have higher seropositivity rates. Therefore, the findings may not reflect the seroprevalence at different stages of the outbreak, and caution should be exercised when comparing results from different settings.

In the "Discussion" section, the limitations can be addressed in the context of the study's objectives, methodology, and potential biases. It is important to acknowledge any potential sources of bias, such as sampling bias or recall bias, and discuss how these limitations may have influenced the results. Additionally, the limitations can be discussed in relation to the broader implications of the study and its contribution to the existing literature.

Reviewer 2 Report

Comments and Suggestions for Authors

In the present manuscript, the authors analyzed the seroprevalence of COVID-19 and factors associated with seropositivity in the Eswatini population. The topic is interesting. Please see some comments below.

General comments

Please correct "seroepidemiology", "COVID-19", and "seropositivity" throughout the manuscript. The number of decimal places should be used consistently. 

Specific comments that should be corrected/rephrased

Abstract

Line 21: ... of the past infections ...

Line 22: ... estimating the extent of ...

Line 28: ... hospitalizations before and after the ...

Lines 29-30: We evaluated the immunoglobulin G (IgG) seropositivity based on either anti-nucleocapsid (N) or anti-spike (S) antigens.

Line 37: ... previous positive SARS-CoV-2 NAAT ...

Line 41: Please explain the abbreviation IFR

Line 42: ... report the decoupling ...

Line 45: ... and before the onset ...

Line 47: ... to the decoupling of ...

Background

Line 68: ... variants have been

Line 74: ... and before the onset of ...

Methods

Line 80: ... the median age ...

Line 98: ... the Our World Data ...

Line 109: DBS sample.

Line 112: ... positivity to determine the seroprevalence.

Line 119: ... at the sub-region level.

Line 126: age groups

Line 139: ... the incidence rate ...

Line 145: ... COVID-19 case numbers ...

Line 146: ... imputing the number ...

Line 157: ... in the writing ...

Results

Lines 164-165: This sentence is not clear, please rephrase. I think it should be also added: The SARS-CoV-2 seroprevalence according to the region is presented in Table 1.

Table 1, columns 8 and 9: the title is not clear. should it be seroprevalence anti-N and anti-S IgG?

Table 2, column 1: please correct "Reported previous COVID-19 test: Never tested

Lines 202-206: The anti-N seropositivity was presented. Why the anti-S seropositivity was not presented? Please add.

Line 221: ... had a higher risk of ...

Line 231:  ... regions, the rate of ... in the Hhohho region

Line 247: ... and a mortality rate of ...

Line 253: The large variance ...

Line 262:  ... reported before the onset ...

Line 267:  ... for waves 1-4 ...

Discussion

Line 291: population-based

Line 292: ... Eswatini before the onset ...

Line 299: household

Line 300: 13-month

Line 309: with a population ...

Line 313: ... highlights the challenges ...

Line 317: The sentence "Our study reports 639,475 SARS-CoV-2 cases compared to 25,048 reported cases" is not clear.

Line 324: ... using RT-PCR positive ...

Line 333: ... cases and deaths ...

Lines 339-340: ... COVID-19 deaths, ...

Line 345: ... of a high number ...

Line 348: The relatively high ...

Line 360: ... and a similar increase ...

Line 379-380: (median 22 years ...)

Line 383: overestimated seroprevalence.

Line 392: ... of infection has taken place in Eswatini before the onset ...

Line 399: infection-induced

Since no confirmatory virus neutralization test was performed (anti-N antibodies may be cross-reactive), this limitation should be added in the Discussion section.

References

References should be checked and corrected according to the propositions of the journal.

For example ref. 12: volume and page numbers are missing.

In some references, the full name of the journal was used and in others, an abbreviation was used.

Some preprints have been published.

For example, ref 31. Tarke A, Coelho CH, Zhang Z, Dan JM, Yu ED, Methot N, Bloom NI, Goodwin B, Phillips E, Mallal S, Sidney J, Filaci G, Weiskopf D, da Silva Antunes R, Crotty S, Grifoni A, Sette A. SARS-CoV-2 vaccination induces immunological T cell memory able to cross-recognize variants from Alpha to Omicron. Cell. 2022 Mar 3;185(5):847-859.e11. doi: 10.1016/j.cell.2022.01.015.

Reviewer 3 Report

Comments and Suggestions for Authors

The reviewed manuscript describes the detection of antibodies against of SARS-CoV-2 in vaccinated and unvaccinated individuals of Eswatini. The work addresses a research gap of the seroprevalence of SARS-CoV-2 in Africa. The incidence data obtained in this study is many times higher than the official data, which explained the unusually high mortality-to-morbidity ratio in official reports. The work contains valuable information, but its design needs improvement. The tables are completely unreadable. I propose to place the tables in their present form in the “supplemental materials”, and to provide an abbreviated version in the article, without confidence intervals. Shorten the headings above the columns and provide explanations below the table.
